# Nano-PSO Administration Attenuates Cognitive and Neuronal Deficits Resulting from Traumatic Brain Injury

**DOI:** 10.3390/molecules27092725

**Published:** 2022-04-23

**Authors:** Doaa Qubty, Kati Frid, Meirav Har-Even, Vardit Rubovitch, Ruth Gabizon, Chaim G Pick

**Affiliations:** 1Department of Anatomy and Anthropology, Sackler Faculty of Medicine, Tel Aviv University, Tel Aviv 6997801, Israel; doaaqubty@mail.tau.ac.il (D.Q.); 5meirav5@gmail.com (M.H.-E.); rubovitc@tauex.tau.ac.il (V.R.); 2Sagol School of Neuroscience, Tel Aviv University, Tel Aviv 6997801, Israel; 3The Agnes Ginges Center for Human Neurogenetics, Department of Neurology, Hadassah University Hospital, Medical School, The Hebrew University, Jerusalem 91120, Israel; kati.frid@gmail.com (K.F.); gabizonr@gmail.com (R.G.); 4Sylvan Adams Sports Institute, Tel Aviv University, Tel Aviv 6997801, Israel; 5The Dr. Miriam and Sheldon G. Adelson Chair and Center for the Biology of Addictive Diseases, Tel Aviv University, Tel Aviv 6997801, Israel

**Keywords:** TBI, Nano-PSO, neuroinflammation, oxidative stress, neurodegeneration

## Abstract

Traumatic Brain Injury (TBI), is one of the most common causes of neurological damage in young populations. It is widely considered as a risk factor for neurodegenerative diseases, such as Alzheimer’s disease (AD) and Parkinson’s (PD) disease. These diseases are characterized in part by the accumulation of disease-specific misfolded proteins and share common pathological features, such as neuronal death, as well as inflammatory and oxidative damage. Nano formulation of Pomegranate seed oil [Nano-PSO (Granagard ^TM^)] has been shown to target its active ingredient to the brain and thereafter inhibit memory decline and neuronal death in mice models of AD and genetic Creutzfeldt Jacob disease. In this study, we show that administration of Nano-PSO to mice before or after TBI application prevents cognitive and behavioral decline. In addition, immuno-histochemical staining of the brain indicates that preventive Nano-PSO treatment significantly decreased neuronal death, reduced gliosis and prevented mitochondrial damage in the affected cells. Finally, we examined levels of Sirtuin1 (SIRT1) and Synaptophysin (SYP) in the cortex using Western blotting. Nano-PSO consumption led to higher levels of SIRT1 and SYP protein postinjury. Taken together, our results indicate that Nano-PSO, as a natural brain-targeted antioxidant, can prevent part of TBI-induced damage.

## 1. Introduction

Traumatic Brain Injury (TBI) is a profound public health concern and major cause of disability and death in young adults and elderly populations. It is considered one of the most common neurological injuries that causes morbidity and mortality and affects more than 10 million people worldwide each year [1]. Since head injuries do not always require seeking proper medical care, these data are likely even higher. The main causes of TBI include falls, road accidents, assaults and sports injuries [2]. TBI is common among men and especially frequent in adolescents and young adults following road accidents and alcohol-related injuries. It is also common in the elderly due to their increased chance of falls [3].

People who have had TBI may suffer from behavioral, cognitive, and emotional impairments, both short-term and long-term, depending on the severity of the injury [3]. Patients exposed to TBI can emerge unscathed but can also be exposed to injuries that require prolonged hospitalization and even end in death. About 40% of cases of severe injury post-TBI will end in death, with survivors suffering from various neurological disorders including epilepsy, dementia, and neurodegenerative diseases [3,4]. Moderate injury causes morphological changes similar to those of severe injury, but they are less severe, and the cognitive damages are mostly milder than in severe injury [5]. In addition, there is well-established evidence that TBI is an important risk factor for the development of neurodegenerative diseases such as Alzheimer’s and Parkinson’s [6,7,8].

Oxidative stress is one of TBI’s potential secondary injuries, which also includes excitotoxicity, inflammation, apoptosis, and mitochondrial dysfunction. Mitochondrial dysfunction is often found in patients with neurodegenerative diseases such as Parkinson and Alzheimer’s [9,10,11,12]. Mitochondrial dysfunction can lead to the unregulated overproduction of the reactive oxygen species (ROS), which is a frequent indicator of TBI secondary injury [13]. The human body produces endogenous antioxidants to offset or inhibit potential increases in ROS production. When TBI leads to an increase in ROS production, the body’s natural antioxidant levels are overwhelmed often causing lipid peroxidation and DNA and protein damage [14]. This can lead to either an overage of ROS or low antioxidant levels, a condition known as oxidative stress that can have devastating and often toxic effects, causing the death of neuronal cells and potentially pathogenesis [15,16,17,18,19].

TBI has no proven treatment for lessening of damage or even prevention. For this reason, it is of paramount importance to seek a better understanding of the pathological mechanisms that follow injury to determine ideal treatment protocols.

Nano-PSO (Granagard ^TM^), a nano-formulation of pomegranate seed oil comprising high levels of punicic acid, the strongest natural antioxidant, was shown to ameliorate neurodegeneration signs, including memory and cell death in several models of neurodegenerative diseases [20,21,22]. In the transgenic mice model for genetic CJD, long-term administration of Nano-PSO resulted in delay of disease progression and reduced cell death. In the 5XFAD mice model for AD, Nano-PSO administration prevented memory impairment and reduced the accumulation of A-beta, both intracellularly and extracellularly. In all models Nano-PSO prevented a reduction in mitochondrial activity, probably by its strong and brain-targeted antioxidant function. In humans, Nano-PSO was recently shown to increase memory in multiple sclerosis patients [23].

In the present study, we show that the administration of Nano-PSO, a dietary supplement both before or after TBI in mice, mitigates the impairments of visual and spatial memory. It also reduces neuronal loss in the temporal cortex and dentate gyrus and reduces reactive astrocyte intensity in the temporal cortex of injured mice brains. Moreover, it restores the production of cytochrome C oxidase (COX), a mitochondrial enzyme that is reported to be affected in brain cells during neurodegeneration [24]. Furthermore, it plays a role in the respiratory chain and is responsible for the reduction of up to 95% of the oxygen taken up by the cells, including neural cells [24].

## 2. Results

### 2.1. Administration of Nano-PSO to Mice before and after TBI Prevents Their Cognitive Deterioration

Following TBI exposure and treatment with Nano-PSO either as a preventive or as a post-TBI treatment (see Figure 1 and methods for scheme of the experiments), male ICR mice were evaluated using the elevated plus maze at two weeks post-TBI among separate cohorts. We evaluated anxiety-like behavior of the mice by recording the time spent in the open arms of the maze and the total number of entrances to each arm, respectively. There were no differences in the time spent in the open arms between all groups at both treatment approaches tested, indicating that anxiety-like behavior of the mice was not affected either by the injury or by treatment with Nano-PSO (results not shown).

To assess the effects of both of the treatment approaches (pre- or post-TBI) of Nano-PSO consumption on memory formation, the novel object recognition (NOR) and Y-maze paradigms were performed on different groups of mice at two weeks postinjury. The NOR paradigm is consistently reported to be impaired by TBI [25]. As expected, TBI mice suffered from significant visual memory deficits and spent less time near the novel object, compared to all the other groups (*p* < 0.01, *p* < 0.001, Figure 2A,B). The deficit was ameliorated by Nano-PSO in both of the treatment approaches. Next, we performed the Y-maze test to evaluate the spatial memory as previously described [26]. At both treatment approaches, TBI mice experienced spatial memory impairment compared to untreated control mice (*p* < 0.001, Figure 3A,B). When mice consumed the treatment by Nano-PSO, spatial memory impairment was ameliorated in both treatment approaches (*p* < 0.001).

### 2.2. Nano-PSO Treatment Prevents Neuronal Death in TBI Mice

Following the behavioral experiments, which found that Nano-PSO can prevent cognitive damage resulting from brain injury, we looked into the levels of TBI brain damage in the presence or absence of Nano-PSO on cellular levels by immunocytochemistry. Mice were terminated and their brains harvested three weeks post injury, at which time further biochemical changes were found [25,27]. The changes were examined in the temporal cortex, CA3 region of the hippocampus, and dentate gyrus on both sides of the brain. The results from both sides of the brain were combined into one value for each mouse. Figure 4A,B show the immunostaining of these brain areas with anti-NeuN antibody, a marker of mature neurons. The results how that the number of neurons was significantly reduced following TBI, but such reduction in neurons was prevented if Nano-PSO was administrated before TBI or post-TBI and until mice were terminated (three weeks later). TBI induced a decline in the neuronal counts within the cortex and dentate gyrus compared to all other groups (** *p* < 0.01, *** *p* < 0.001), indicating a significant neuronal loss at three weeks following the injury (Figure 4A,B). Both of the treatment approaches preserved neuronal counts in the cortex and dentate gyrus. However, preventive treatment further mitigated the neuronal loss in these areas (** *p* < 0.01, *** *p* < 0.001) (Figure 4A) compared to the treatment after TBI (* *p* < 0.05) (Figure 4B).

### 2.3. Administration of Nano-PSO Reduces Inflammation Caused by TBI

Evaluation of neuro-inflammatory processes occurring subsequent to TBI exposure was obtained by immune-histochemical staining using GFAP antibody which labels reactive astrocytes [28,29]. The staining was performed to examine whether there were changes in the expression level of the astrocytes TBI three weeks after injury and whether a preventive or post-TBI treatment with Nano-PSO affected these changes. Brain slices obtained from the temporal cortex were studied three weeks postinjury. As can be seen in Figure 5, TBI exposure induced a significant elevation in astrocyte in the cortex (*p* < 0.001, *p* < 0.01), respectively (Figure 5). When Nano-PSO was administrated as a preventive or as a treatment post-TBI, it led to a significant reduction in the activated astrocytes in the cortex, *p* < 0.001, *p* < 0.05, respectively (Figure 5). These results demonstrate that due to TBI, GFAP levels are elevated while administration of Nano-PSO reduces GFAP to one close to that of healthy control mice.

### 2.4. Nano-PSO Restores COX IV1 Activity following TB1

It was shown previously that in spontaneous/genetic models of neurodegenerative diseases, such as AD and CJD, the expression of COX IV1, an important mitochondrial enzyme for energy production in cells, is drastically reduced and replaced by COX IV2, which can function under high levels of ROS [24,30]. However, this is a temporary effect until such levels of ROS finally result in apoptosis and cell death. Nano-PSO administration reduces ROS levels and restores COX IV1expression [30]. Therefore, in this work, we tested if this is also the case for TBI, an outside injury caused to normal mice. To this effect, we immunostained brain slices as described above with COX IV1 antibody. Figure 6 shows that, as was the case for the genetic diseases, TBI also causes a reduction in the levels of COX IV1 even three weeks after infliction of the damage in brains. In this case, and as shown before [30,31], the administration of Nano-PSO increases COX IV1 levels both in the control and in the pre- and post-TBI mice. Since an increase in COX IV1 levels reflects the oxidative status of cells, we conclude that Nano-PSO can also reduce ROS levels in control mice and not only in the TBI-affected mice.

### 2.5. Nano-PSO Elevates SIRT1 and SYP Levels Post-TBI

The Immunoblot of the total levels of SIRT1, a marker for NAD-dependent deacetylase sirtuin-1 in the cortex, was examined for all mice three weeks post-TBI (Figure 7). A reduction of SIRT1 levels was found. These results are in line with a previous study with a mild TBI model from our lab [32]. However, this reduction was prevented with Nano-PSO when it was administrated immediately for several weeks after the event.

Figure 7B shows an immunoblot of total levels of synaptophysin (SYP), a marker for synaptic structures, from brain homogenates of control mice treated or untreated with Nano-PSO as well as from TBI or post-TBI-treated mice. The results show that, concomitant with the neuronal death shown in Figure 4A,B, synaptic damage was also caused by TBI. However, this damage can be prevented if a strong and brain-directed antioxidant such as Nano-PSO is administrated immediately for several weeks after infliction of the damage. It has been shown before that most of the affected neurons did not die immediately after the injury but in the days after due to the extensive oxidative stress caused by TBI [33]. Therefore, immediate antioxidant treatment can be efficient enough to prevent extensive damage in the long run.

Below each graph, a representative image of the levels of SIRT1/SYP and the household protein tubulin (TUB) in the right cortex among all the groups is presented.

## 3. Discussion

In this study we showed that mice suffered significant visual and spatial memory deficits following TBI induction. These deficits were ameliorated by treatment with Nano-PSO (Granagard^TM^) either as a preventive care or as a post-TBI treatment. Cognitive deficits were most probably related to neuronal death, gliosis and mitochondrial damage. All these pathologies were reduced by Nano-PSO treatment, both as preventive care or as postinjury treatment. Nano-PSO also prevented a reduction in SIRT1 levels following TBI and increased levels of SYP compared to injured, untreated mice.

These results indicate that Nano-PSO plays a neuroprotective role following TBI in mice brains. PSO comprises high levels of punicic Acid (PA), a conjugated unsaturated fatty acid, While the oil in its natural form has been shown to reduce disease burden in other models only when given at very high doses, a superior clinical effect was achieved when an amount equivalent to just 1% was administered in the form of emulsified Nano droplets of PSO [20]. Application of natural supplements is often prescribed as a preventive measure for neurodegenerative diseases. Therefore, in this study we administrated Nano-PSO not only as an after TBI treatment, but also as a preventive measure.

The pathophysiology of TBI is complex and characterized especially by neuronal cell death and axonal damage, induced primarily by the direct physical impact itself and secondarily by oxidative stress, neuro-inflammation, mitochondrial dysfunction, DNA damage and other pathological processes [34,35]. Previous studies have shown that the active ingredient of Nano-PSO is CLA, conjugated linoleic acid, a metabolite of PA, and that in this formulation it can be targeted to the brain. Once in the brain, CLA can inhibit memory decline and neuronal death in mice models of AD and genetic Creutzfeldt Jacob disease (gCJD), respectively [31,36]. Our study, in addition to the previous once, can demonstrate that Nano-PSO has the potential to serve as an antioxidant neuroprotective agent pre- and post-TBI with a wide array of neuronal and clinical benefits.

Previous studies have shown that several biochemical changes take place at different points in time following [37,38,39]. In our examination of the cortex and hippocampal regions, a significant elevation in neurodegenerative markers was observed across all regions. TBI induces neuro-inflammation, which is reflected in a higher expression of astrocytes in the temporal cortex, a cell population that plays a significant role in neuro-inflammation [29,40] and that may contribute to the collapse of protection mechanisms. In addition, the extent of TBI-induced neuronal death strongly correlates with the development of cognitive deficits [41,42]. TBI leads to rapid neuronal cell death resulting from direct contusion, followed by progressive secondary cell death in surrounding tissues [42]. Polyunsaturated fatty acids, known as PUFA, have been shown to exert potent protection against oxidative stress, glutamate-induced excitotoxicity, apoptotic cell death, and inflammation [43,44]. Long-term dietary supplementation with fatty acids has been shown to effectively reduce neuronal loss post-TBI [45]. Building upon those previous findings, the current study revealed that Nano-PSO consumption, whether pre- or post-TBI, provides significant protection against neuronal loss in the cerebral cortex and dentate gyrus. In addition to Nano-PSO, a similar polyunsaturated fatty acid treatment has been investigated for its potential to prevent elevation in neurodegenerative processes, with results demonstrating that it attenuates TBI-induced neuronal apoptosis [46].

This study also provides insights on TBI and mitochondrial malfunction. Mitochondrial malfunction is a hallmark of neurodegenerative diseases such as AD, PD, HD, and many others [10,11,47] as well as TBI [14]. The loss of mitochondrial function caused by TBI disrupts normal brain functioning and reduces neuronal repair [48]. Under oxidative stress and degenerative conditions, the COX IV1 activity is replaced by COX IV2 [49], resulting in an increase of total COX activity and ATP production. Mitochondrial abnormalities and compensatory mechanisms under oxidative stress further reduce the expression of COX IV1 [31]. This study showed that Nano-PSO application elevated the expression of COX IV1 and attenuated the mitochondrial effects of TBI. Nano-PSO is mainly an antioxidant and as such may also have an effect on wild-type animals, since while oxidation in the brain and other organs is significantly elevated in pathological neurodegenerative conditions, it is also present in basal conditions. Indeed, we have shown recently that Nano-PSO can be considered a general anti-aging reagent, since it may correct a variety of aging hallmarks, similar to Metformin [50].

Finally, to understand how Nano-PSO regulated neuroprotective protein expression in our moderate TBI model, we examined the levels of SIRT1 and synaptophysin (SYP) in the cortex of mice treated with Nano-PSO post-TBI through Western blot analysis. Mice who received the treatment preinjury received significantly more Nano-PSO than those treated postinjury. Since treatment both before and after injury eliminated visual and spatial memory deficits as well as other analyzed factors, we only performed Western blot analysis on mice treated postinjury.

Overexpression of SIRT1 has been shown to increase cell viability and reduce inflammatory cytokines [51]. SIRT1 is also involved in the aging process and serves as a factor improving life longevity in animal models [52,53,54]. Furthermore, SIRT1 serves as a deacetylase for numerous proteins involved in several cellular pathways, including stress response and apoptosis [55]. This pathway appears to be involved in the modulation of synaptic plasticity and memory formation [56,57]. Finally, SIRT1 plays a significant role in induced neuroprotection following fatty acid intake [46]. In our model, injured mice demonstrated a significant impairment in visual and spatial memory, which suggests SIRT1 involvement in the mechanism contributing to cognitive impairment. The results showed that administration of Nano-PSO, which is comprised of fatty acids, led to a recuperation of lost SIRT1 and mitigated the cognitive impairments induced by TBI. In addition, in prior research by our lab showed that SIRT1 reduction in the cortex and hippocampus was accompanied by cognitive impairments in a mouse model of mild TBI [32].

Numerous studies involving animal models of TBI have described changes in synaptic protein (SYP) levels following TBI [58,59,60,61]. These studies describe both reductions and increases in SYP protein following TBI in the cortical regions from day 1 to day 30. In our moderate closed head TBI model, we detected reductions in protein labelling in neurons of the pre-synaptic protein SYP. This reduction is in line with prior research from our lab, which was conducted in a blast TBI model [62]. In the current study, this damage was prevented by the administration of Nano-PSO immediately post-TBI. Nano-PSO elevated SYP levels compared to untreated, injured mice.

Synaptic function preservation is associated with the elevation of synaptophysin protein expression [63]. Such expression is usually accompanied by an increase in astrocytic glial fibrillary acidic protein (GFAP) expression [64]. The injured mice in this study demonstrated elevated GFAP expression. Past research shows that most of the affected neurons do not die immediately after the injury but in the days thereafter due to the extensive oxidative stress caused [33]. Therefore, immediate treatment with Nano-PSO can be efficient enough to prevent extensive damage brought on by GFAP in the long run.

Overall, Nano-PSO treatment post-TBI attenuated the injury-induced expression of SIRT1 and elevated SYP levels. This likely contributes to the manifested neuroprotective effects of Nano-PSO. Indeed, both of these proteins are involved in the mechanisms and cellular pathways that Nano-PSO uses to elevate neuronal survival, increase COX IV1 levels, and restore mitochondrial activity to better normal levels. Taken together, these mechanisms led to a full reversal of cognitive impairments.

This study demonstrates that Nano-PSO treatment, administered either as preventive care or as treatment postinjury, is effective in ameliorating cognitive impairments and secondary damages that occur as a consequence of moderate TBI. Our findings contribute to and expand on a growing body of evidence demonstrating the therapeutic effects of Nano-PSO, giving practitioners a natural treatment option for TBI patients or as preventive care for those with a high risk of experiencing a brain injury. Future studies in our collaborative group will focus on the effect of Nano-PSO on MMP-2 and MMP-9. Recent literature discussed the ability of natural substances of terrestrial origin to modify the levels of MMP2/9 [65]. MMP2/9 are targets in neurodegenerative diseases. In particular, they play an important role in the case of neurodegeneration and are able to trigger several neuro-inflammatory and neurodegenerative pathways [65].

## 4. Materials and Methods

### 4.1. Animals Use in Experiments

Mice of adult age, approximately 6 through 8 weeks, and weighing between 31 and 34 g, were held in cages in groups of 5 with food and water, with a 12 h dark/light cycle and a consistent temperature of 22 ± 0.5 °C. Experiments were carried out only during light. Mice only participated in one experiment each, and extensive efforts were made to reduce any possible suffering. The experiment protocol (01-19-059) was approved by the Ethics Committee of the Sackler Faculty of Medicine, Tel Aviv, Israel in accordance with animal experiment guidelines from the National Institutes of Health (DHEW publication 85-23, revised, 1995).

#### 4.1.1. Nano-PSO

Preparation of Nano-PSO self-emulsifying formulation was as previously described [21] and is defined in patent No. 14/523,408.

#### 4.1.2. Administration of Nano-PSO to Mice

In this study, two different protocols for the administration of Nano-PSO were conducted, a preventive treatment a week prior to TBI or a treatment that was given directly after TBI induction. Three days were given to mice upon arrival to the animal facility for well-being and habituation; Nano-PSO was administrated to the two groups of mice in each experiment. First protocol; a week prior to TBI induction and for three weeks after the injury, mice received 1.6% oil/mL by adding a self-emulsion formulation into drinking water as previously described [21]. In the second protocol, the same amount of Nano-PSO was given directly after TBI and for three weeks after.

#### 4.1.3. Traumatic Brain Injury Model

A weight drop head trauma device was used to induce brain injury [66,67]. It is made up of a metal pipe with a length of 80 cm and a diameter of 13 mm. Isoflurane inhalation was used before injury induction in order to lightly anesthetize the mice. The mice were then placed below the device on top of a sponge to keep its head in place. A weight weighing 70 g was dropped at the top of the tube, free falling and hitting the mouse’s right temporal lobe between the ear and the eye. Directly following induction of the injury, mice were placed back in their cages to recover and wait for follow up. Mice in the control group received similar treatment. They receive anesthesia with isoflurane, and then were placed under the weight drop device without being exposed to the weight drop itself. This ensured that the anesthesia did not induce any deficits. Following this, all animals were placed back in their cages for recovery. The TBI mice and control group were indistinguishable after recovering. We selected animal numbers for individual studies by assessing variance in data from previous studies by our lab.

#### 4.1.4. Experimental Procedure

The timeline of administration of Nano-PSO dosage was divided into two treatment approaches. Preventive treatment, which started three days post-arrival to the facility and a week prior to TBI induction is shown in Figure 1, A second treatment approach is shown in Figure 1B. In this approach, the administration of Nano-PSO started only post-TBI induction, and mice received the treatment until the brain harvest. The behavioral tests were performed two weeks following the injury in separate groups of animals (Figure 1A,B). Immuno-histochemical staining assessments and Western blot were performed on brains that were collected postinjury.

### 4.2. Behavioral Tests of Cognitive Function

#### 4.2.1. Novel Object Recognition

To investigate the mice’s recognition and visual memory, as described previously, the NOR task was used [68]. The NOR’s effectiveness relies on rodents’ innate tendency to explore novel items around them [69]. This allows researchers to conclude whether mice can distinguish between a novel object and a familiar one. For a period of five minutes, mice were placed one by one in an open-field arena (60 × 20 × 20 cm) made of plexiglass. After 24 h, in a subsequent acquisition phase, two identical objects (A or B) were placed within the arena in symmetrical positions. In the memory recognition portion of the test, one of the objects (either A or B) was replaced by a new one (C). The mice’s behavior was then assessed during the five-minute period. We recorded how much time the mouse spent exploring the familiar object versus the novel object by the preference index: (time spent near the new object minus time spent near the old object)/(time spent near the new object plus time spent near the old object) [69].

#### 4.2.2. Y-Maze

The Y-maze paradigm evaluates spatial memory function, responsiveness while in new environments and spontaneous exploration, as previously described [26]. A three-armed black maze made from plexiglass was used for the Y-maze study, with each arm separated by a measure of 120 degrees. The arms were 8 × 30 × 15 cm and can be told apart only by spatial cues such as a square, circle or triangle. All mice used the same arm as a starting point. Mice were put into the Y-maze in two rounds, separated by 2 min intervals. During the time between rounds, the mice were placed back in their home cages. In the first round, one of the two arms was blocked. In the second, all of the arms were kept open for exploration. We measured the total time that mice explored each arm. We calculated a discrimination preference index as such: (time in new arm − time in familiar arm)/(time in new arm + time in familiar arm) [69].

#### 4.2.3. Immunohistochemical Studies

Brains were harvested three weeks after injury as previously described [27]. To assess the neuronal survival and changes in the expression of reactive astrocytes, we performed a double staining using neuronal nuclear antigen (NeuN), a marker of mature neurons and rabbit glial fibrillary acidic protein (GFAP), a marker for reactive astrocytes. Sections were incubated for 48 h with a mouse primary antibody that detects NeuN (Millipore, Burlington, MA, USA; MAB377, 1:50 in incubation buffer) and with rabbit glial fibrillary acidic protein (GFAP) primary antibody (Dako, Glostrup, Denmark; Z0334, 1:500). After 48 h of incubation with both of the primary antibodies, sections were washed and incubated with a Cy3 labeled anti-mouse secondary antibody and with donkey anti-rabbit secondary antibody (Abcam, Cambridge, UK; Goat Anti-Mouse IgG H&L (Alexa Fluor^®^ 488 ab150113), and Goat Anti-Rabbit IgG H&L (Alexa Fluor^®^ 488, ab150077), 1:300 in incubation buffer) for an hour at room temperature.

To evaluate the mitochondrial dysfunction that was caused by TBI, we performed a staining with Anti-COX IV antibody. The primary antibody (mAbcam33985, ab33985) was used as a mitochondrial marker. A series of 30-micron sections were incubated for 48 h with rabbit monoclonal antibody raised against a human COX IV1 peptide which in mice recognizes only the COX IV1 isomer (Abcam, ab202554). The sections were then washed and incubated with donkey anti-rabbit secondary antibody (Abcam, Cambridge, UK; Alexa Fluor^®^ 594 ab150064, 1:300 in incubation buffer) for an hour at room temperature.

Sections were then rinsed with PBST and mounted onto 2% gelatin-coated slides. Using a Leica SP5 confocal microscope (Leica, Wetzlar, Germany) sections were observed in a 40 or 60 magnification. Four sections of the brain were used to take six to eight images from the cortex and hippocampus. The images were drawn from both sides of the brain and then analyzed through the Imaris software (Bitplane AG, Zurich, Switzerland), thereby providing an average figure for each region of the brain. Nuclei were labeled with DAPI Fluoromount (Sigma Aldrich, Burlington, MA, USA).

#### 4.2.4. Western Blot Analysis

Three weeks after TBI induction, brain extracts were homogenized and dissociated in a buffer lysis (Tissue Protein extraction Reagent, Pierce San Antonio, TX, USA). Using a Teflon pestle homogenizer, these actions were supplemented with a cocktail (Halt Protease Inhibitor Cocktail, Sigma-Aldrich, Burlington, MA, USA). We added sample buffer to each of the samples and then placed them in storage at a temperature of −20 °C. We then heated samples up to 90 °C for a period of three min. Then 30 μL of each sample was loaded and run on 4–20% Mini-Protean TGX gels (Bio-Rad, Hercules, CA, USA; 456-1094). They were then transferred onto nitrocellulose membranes (Bio-Rad, Hercules, CA, USA; 1704159) using a common transfer system (Trans-Blot Turbo, Bio-Rad, Hercules, CA, USA). Thereafter, the blots were blocked for 1 h at normal room temperature in a Tris-buffered saline that contained 0.01% Tween-20 and 5% BSA. We then incubated the membranes at 4 °C overnight with a primary mouse anti-SIRT1 antibody (diluted 1:500; ab10304) or SYP antibody (diluted 1:5000; ab32594). This was followed by washings with TTBS. We then incubated the membranes at room temperature for one hour with a horseradish peroxidase–conjugated goat anti-mouse/rabbit antibody (Jackson immune Research Laboratories, Inc., West Grove, PA, USA) or (Jackson, Baltimore Pike, West Grove, PA, USA; 111-035-003, 1:10,000). Thereafter, bands were visualized through enhanced chemiluminescence reagents for a period of one minute (enhanced chemiluminescence assay) (Millipore, Billerica, MA, USA) using the Viber Fusion FX7 imaging system (Viber Lourmat, Eberhardzell, Germany). We used ImageJ software to create a densitometry analysis of the detected signal. We verified uniform loading by re-probing and stripping with a mouse primary α-tubulin antibody for a period of thirty minutes at room temperature (Santa Cruz, Dallas, Texas, United States; sc-53030, 1:10,000). This was followed by a conjugated goat anti-mouse secondary antibody (Jackson, Baltimore Pike, West Grove, PA, USA; 115-035-003, 1:10,000). We determined the value of each sample by the ratio of SIRT1 and α-tubulin. We set averages in control values in each membrane at one and calculated all other samples accordingly.

#### 4.2.5. Data Analysis

Results were analyzed using SPSS V 25 software and presented as mean plus or minus SEM. For comparisons between datasets, One-way ANOVA tests were used. When found significant, this was followed by Fisher’s least significant difference (LSD) post hoc analysis. Significant values between means are expressed as * *p* < 0.05, ** *p* < 0.01, *** *p* < 0.001.

## Figures and Tables

**Figure 1 molecules-27-02725-f001:**
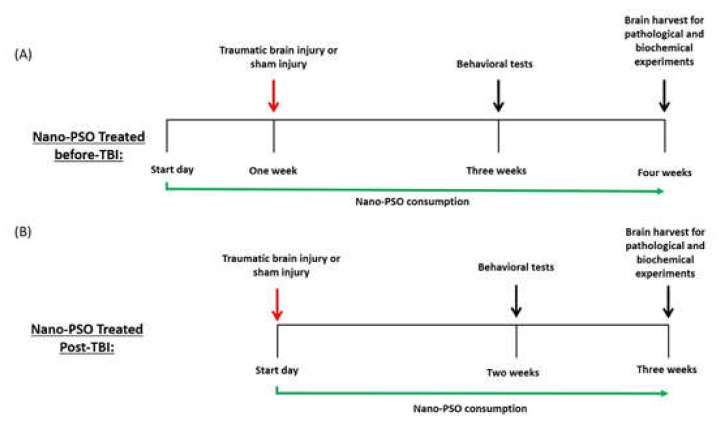
Outline of experiments. (**A**) Animals were treated with Nano-PSO (1.6% oil/mL) or drinking water for one week prior the exposure to TBI; (**B**) Animals were treated with Nano-PSO (1.6% oil/mL) or drinking water immediately after exposure to TBI. Both experiments continued consuming Nano-PSO to the end point of each experiment, except the untreated control or TBI group. Behavioral tests were carried out two weeks following TBI for both experiments in separate cohorts of animals.

**Figure 2 molecules-27-02725-f002:**
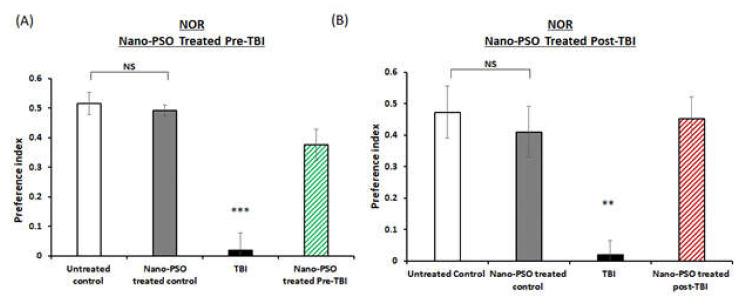
Recognition of a novel object was improved by Nano-PSO treatment in TBI mice. TBI mice treated or untreated were subjected to the novel object recognition task (NOR) at two weeks following TBI as described in the methods: (**A**) mice treated with Nano-PSO before TBI; and (**B**) mice treated with Nano-PSO post-TBI represent preference index of time spent exploring the new object compared to the familiar object. Statistical analysis by One-way ANOVA revealed that TBI animals had a deficit in visual memory compared to all other groups. (**A**: F (3, 33) = 25.390, *p* = 0.000 Fisher’s LSD post hoc, *** *p* < 0.001, *n* = 8–10), (**B**: F (1, 15) = 14.127, *p* = 0.00 Fisher’s LSD post hoc, ** *p* < 0.01, *n* = 7–10). NS = Not significant. Values are presented as mean ± SEM.

**Figure 3 molecules-27-02725-f003:**
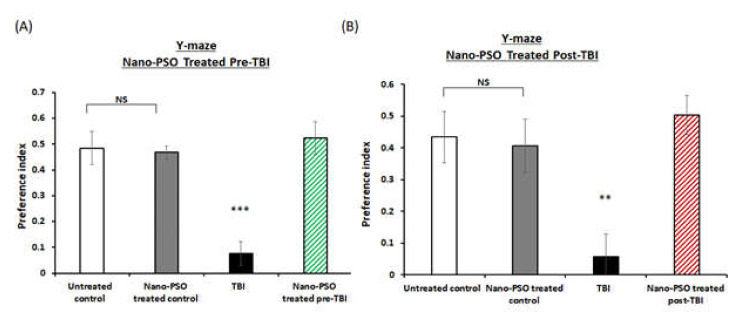
Nano-PSO administration to TBI mice improved spatial memory assessed in the Y-maze test. TBI mice treated or untreated were subjected to the Y maze test at two weeks following TBI as described in the methods: (**A**) mice treated with Nano-PSO before TBI; and (**B**) mice treated with Nano-PSO post-TBI represent preference index of the relative time that mice spent exploring the novel arm compared to the old arm. Statistical analysis by One-way ANOVA revealed that TBI animals had a deficit in spatial memory compared to all other groups. (**A**: F (3, 36) = 16.039, *p* = 0.000 Fisher’s LSD post hoc, *** *p* < 0.001, *n* = 10; **B**: F (1, 15) = 14.127, *p* = 0.00 Fisher’s LSD post hoc, ** *p* < 0.01, *n* = 10). NS = Not significant. Values are presented as mean ± SEM.

**Figure 4 molecules-27-02725-f004:**
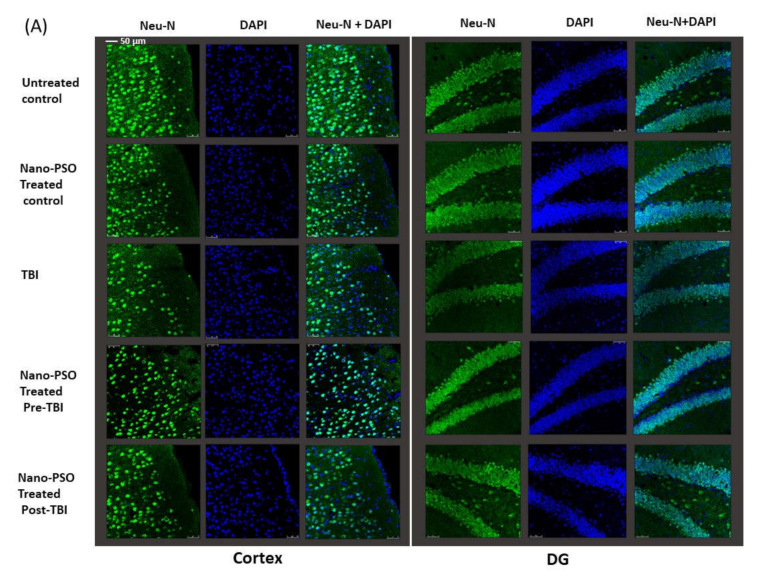
Nano-PSO treatment mitigates neuronal loss in the brain of TBI mice: (**A**) coronal sections through the temporal cortex and dentate gyrus of brains from treated and untreated controls, as well as from untreated TBI, treated with Nano-PSO before TBI and treated with Nano-PSO post-TBI were immunostained for NeuN (green) and DAPI (blue) [scale bar 50 µm]; (**B**) quantitative assessment of NeuN immunostaining for all sections (bregma temporal cortex −1.06 nm and dentate gyrus −1.34 nm) at the end point of each experiment was quantified using One-way ANOVA (Cortex: PRE (F (3, 16) = 15.469, *p* = 0.000 Fisher’s LSD post hoc, *** *p* < 0.001, N = 5); PSOT (F (3, 12) = 8.607, *p* = 0.000 Fisher’s LSD post hoc, * *p* < 0.05 ** *p* < 0.01, *n* = 3–5)), (DG: PRE (F (3, 14) = 6.804, *p* = 0.000 Fisher’s LSD post hoc, ** *p* < 0.01, *n* = 4–5); PSOT (F (3, 13) = 12.580, *p* = 0.000 Fisher’s LSD post hoc, * *p* < 0.05, ** *p* < 0.01, *** *p* < 0.001, *n* = 3–4)). NS = Not significant. Values are presented as mean ± SEM.

**Figure 5 molecules-27-02725-f005:**
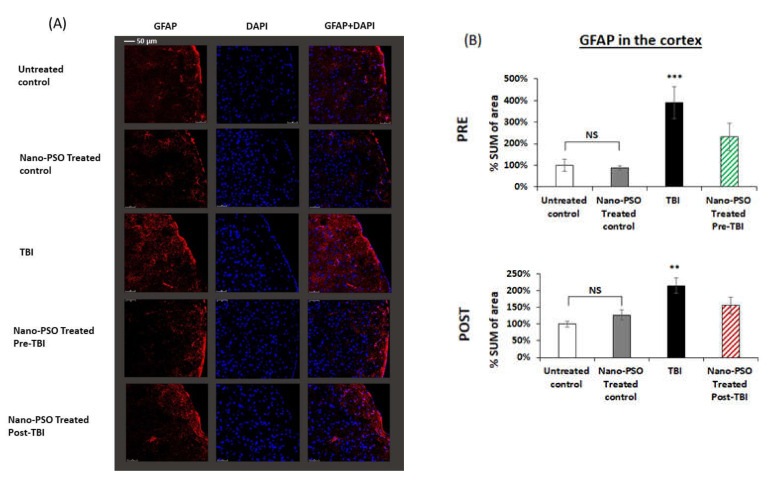
Nano-PSO treatment significantly reduced the elevation of activated astrocytes in the cortex of TBI mice: (**A**) Coronal sections through the temporal cortex of brains from treated and untreated controls, as well as from untreated TBI, treated with Nano-PSO before TBI and treated with Nano-PSO Post-TBI were immunostained for GFAP (red) and DAPI (blue) [scale bar 50 µm]; (**B**) Quantitative assessment of GFAP immunostaining for all sections (bregma temporal cortex -1.06nm) at the end point of each experiment was quantified using One-way ANOVA ((PRE: F (3, 8) = 7.741, *p* = 0.000 Fisher’s LSD post hoc, *** *p* < 0.001, *n* = 3) (POST: F (3, 16) = 6.679, *p* = 0.001 Fisher’s LSD post hoc, ** *p* < 0.01, *n* = 5)). NS = Not significant. Values are presented as mean ± SEM.

**Figure 6 molecules-27-02725-f006:**
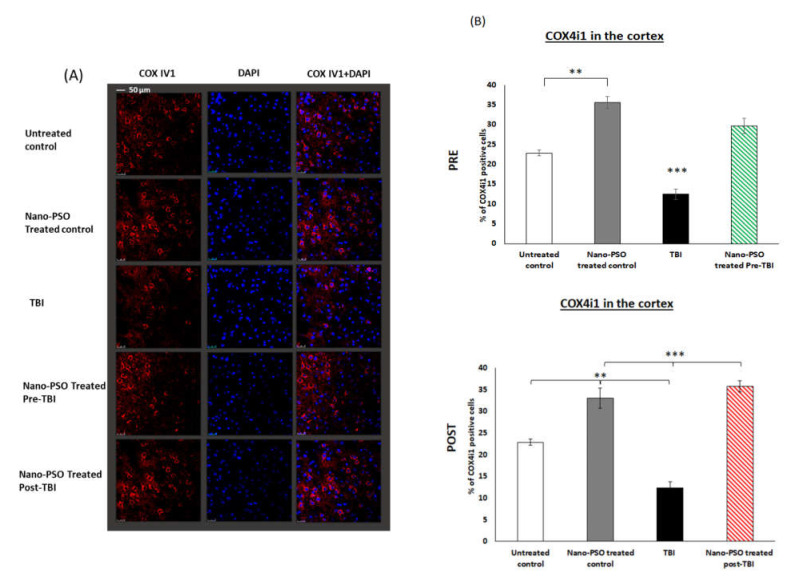
Nano-PSO treatment restored mitochondrial COX IV1 expression in all mice: (**A**) Coronal sections through the temporal cortex of brains from treated and untreated controls, as well as from untreated TBI, treated with Nano-PSO before TBI and treated with Nano-PSO post-TBI were immunostained for COX IV1 (red) and DAPI (blue) [scale bar 50 µm]; (**B**) Quantitative assessment of COX IV1 immunostaining for all sections (bregma temporal cortex −1.06 nm) at the end point of each experiment was quantified by One-way ANOVA (F (4, 12) = 45.101, *p* = 0.000 Fisher’s LSD post hoc, ** *p* < 0.01, *** *p* < 0.001, *n* = 3–4). NS = Not significant. Values are presented as mean ± SEM.

**Figure 7 molecules-27-02725-f007:**
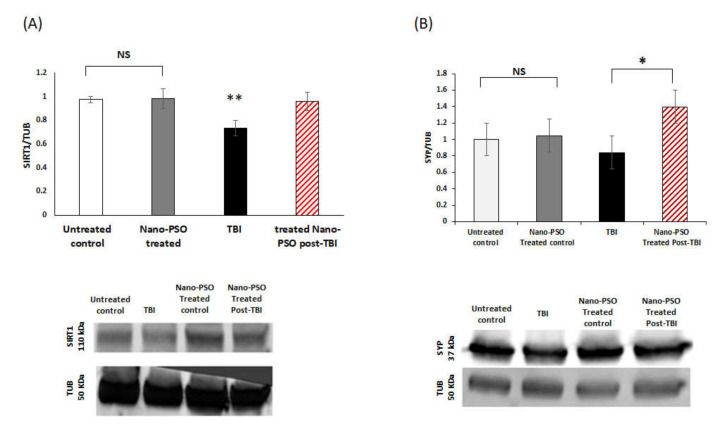
The impact of Nano-PSO on SIRT1/SYP expression in the cortex of TBI mice: (**A**) Brain homogenates from the right cortex of treated and untreated controls, as well as from TBI and Nano-PSO treated post-TBI mice were immunoblotted with SIRT1 and TUB detected at (110 KDa, 50 KDa), respectively. Quantitative analysis of immunoreactive bands of SIRT1 were quantified by Image J and normalized against the control protein TUB. Levels of SIRT1 were significantly reduced 30 days postinjury in the cortex of TBI mice compared to the control. Nano-PSO treatment prevented this reduction ** *p* < 0.01. One-way ANOVA revealed a significant elevation in the expression of SIRT1 in mice that received the treatment postinjury. (F (3, 20) = 1.084, *p* = 0.002, N = 5 Fisher’s LSD post hoc). (**B**) Brain homogenates from the right cortex of treated and untreated controls, as well as from TBI and Nano-PSO treated post-TBI mice were immunoblotted with SYP and TUB detected at (37 KDa, 50 KDa), respectively. Quantitative analysis of immunoreactive bands of SYP were quantified by Image J and normalized against the control protein TUB. One-way ANOVA revealed a significant elevation in expression of SYP in mice that received the treatment post-TBI compared to the TBI group (F (3,20) = 1.084, *p* = 0.379, Fisher’s LSD post hoc * *p* < 0.5, *n* = 4–5). NS = Not significant. Values are presented as mean ± SEM.

## Data Availability

Not applicable.

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
