# Peer review of "Nano-PSO Administration Attenuates Cognitive and Neuronal Deficits Resulting from Traumatic Brain Injury"

_molecules, 2022, doi:10.3390/molecules27092725_

Round 1

Reviewer 1 Report

Nano PSO administration attenuates cognitive and neuronal deficits resulting from traumatic brain injury

The manuscript is very interesting. However, in my opinion, the experimental data lack the comparison with the main component of Pomegranate seed oil Nano-PSO (Granagard TM) punicic acid (PA) and its metabolite CLA.

The Authors should try to explain whether the effect evaluated in vivo by the formulation (Nano PSO) is due to PA acid or its metabolite.

Furthermore, the authors in the introduction should take into account the recent literature on natural substances of terrestrial origin that are able to modify the levels of MMP2 / 9.  MMP2 / 9 are target in neurodegenerative diseases, in particular, they play an important role in the case of neurodegeneration or neuroregeneration.

In my opinion, the manuscript can be accepted if the Authors can compare in in vitro cellular experiment the neuroprotective effect the of Nano-PSO (Granagard TM) and PA and its metabolite CLA.

Author Response

Dear Reviewer, 

Thank you for taking the time and reading my work we appreciate all your comments and I have made changes accordingly. Here is our answers to your comments: 

  • The Authors should try to explain whether the effect evaluated in vivo by the formulation (Nano PSO) is due to PA acid or its metabolite.

The comparison between Nano-PSO and pomegranate seed oil (PSO) before the formation of the nanoformulation was performed in the first manuscripts for this compound on other neurological models. We showed, both for EAE and for familial CJD that Nano-PSO was significantly more active than PSO by itself in the same concentrations  (1, 2).  In an additional paper describing the effect of Nano-PSO on the 5FAD Alzheimer's mice model, we also show that the compound reaching the brain is indeed CLA and not PA (3), indicating this is indeed the active compound. In an additional set of experiments performed in the Gabizon laboratory, the activity of CLA by itself as well as Nano-CLA were tested directly. Our unpublished results show that Nano-CLA was active as Nano-PSO but at some points showed some levels of toxicity. All this taken together indicates the active compound is CLA in the brain, and that the best way to achieve this goal is by a Nano-PSO compound.

  • Furthermore, the authors in the introduction should take into account the recent literature on natural substances of terrestrial origin that are able to modify the levels of MMP2 / 9.  MMP2 / 9 are target in neurodegenerative diseases, in particular, they play an important role in the case of neurodegeneration or neuroregeneration.

Indeed the new papers about MMP2 and neurodegeneration are very interesting, and we have now mentioned this in the discussion. It will be interesting in the future to test the effect of Nano-PSO on these proteins in some of our models. (4)

  • In my opinion, the manuscript can be accepted if the Authors can compare in invitro cellular experiment the neuroprotective effect the of Nano-PSO (Granagard TM) and PA and its metabolite CLA.

As stated above, we have shown in vivo that CLA in brain is the active compound.  As for in-vitro cell culture experiments, these will not allow delivery to the organ of interest therefore there were irrelevant to us. 

Reviewer 2 Report

In general, the manuscript is interesting and contributes to the field. Unfortunately, it has a series of problems when explaining the methodology used, and a series of errors are made when citing the bibliography.

A point of significant concern is the methodological explanation.

Figure 2 does not explain how the preference index is obtained. This also does not appear in the methods. What is appropriately done in figure 3 (Y-maze).

The analysis of statistical differences is erratic. For example, in some figures (4B post effect), a comparison is made between each of the experimental conditions, which seems appropriate. But in the rest of the figures, only some conditions are compared, even including those that indicate "ns", but it is not explained why the other comparisons are included (or not).

The images do not include a scale bar, although it is even indicated that one is used (legend figure 5).

Cited manuscripts need to be checked carefully in relation to what the text says. An example includes citation 23, which is not related to Nano-PSO. And citation 68 seems to be more appropriate than 67. Please check.

In the images of the original blots, it is not clear from which part of them the selected parts for figure 7 were obtained. If 4 lanes are used in a row as indicated in the legend of the original figure, it does not seem necessary to cut each lane to assemble the final figure, as occurred in figure 7 (SIRT1).

The discussion is appropriate, but the conclusion that the effect on the mitochondria is eliminated is too risky.

The effect of Nano-PSO on the untreated control in figure 6 (CoxIV-1) remains to be discussed in depth.

Why is only the "post" effect analyzed in figure 7?

Some specific items:

Include a reference of the analysis that is carried out with the GFAP mark.

Include molecular weights in figure 7

Author Response

Dear Reviewer,

Thank you for taking the time and reading my work we appreciate all your comments and I have made changes accordingly. Here are our answers to your comments in the attached word document. 

Thank you.

Round 2

Reviewer 2 Report

The authors only addressed some of the elements requested.
